# Zipf's laws of meaning in Catalan

**Neus Català**[1][☋], **Jaume Baixeries**[ID][2‡], **Ramon Ferrer-i-Cancho**[2☋], **Lluís Padró**[1‡], **Antoni Hernández-Fernández**[ID][2,3☋]*

**1** TALP Research Center, Computer Science Departament, Universitat Politècnica de Catalunya, Barcelona, Catalonia, Spain, **2** LARCA Research Group, Complexity and Quantitative Linguistics Laboratory, Computer Science Departament, Universitat Politècnica de Catalunya, Barcelona, Catalonia, Spain, **3** Societat Catalana de Tecnologia, Secció de Ciències i Tecnologia, Institut d'Estudis Catalans - Catalan Studies Institute, Barcelona, Catalonia, Spain

☋ These authors contributed equally to this work.
‡ These authors also contributed equally to this work.
* antonio.hernandez@upc.edu

**Data Availability Statement:** The links to the datasets are contained in the paper. These are: The CTILC corpus is available at https://ctilc.iec.cat/ The Glissando Corpus is available at ELRA Catalogue of Language Resources http://catalog.elra.info/en-us/

## Abstract

In his pioneering research, G. K. Zipf formulated a couple of statistical laws on the relationship between the frequency of a word with its number of meanings: the law of meaning distribution, relating the frequency of a word and its frequency rank, and the meaning-frequency law, relating the frequency of a word with its number of meanings. Although these laws were formulated more than half a century ago, they have been only investigated in a few languages. Here we present the first study of these laws in Catalan. We verify these laws in Catalan via the relationship among their exponents and that of the rank-frequency law. We present a new protocol for the analysis of these Zipfian laws that can be extended to other languages. We report the first evidence of two marked regimes for these laws in written language and speech, paralleling the two regimes in Zipf's rank-frequency law in large multi-author corpora discovered in early 2000s. Finally, the implications of these two regimes will be discussed.

## Introduction

During the 1st half of the last century, G. K. Zipf carried out a vast investigation of statistical regularities of languages [1–3], that lead to the formulation of linguistic laws [4]. Among them, a subset has received very little attention: laws that relate the frequency of a word with its number of meanings in two ways. On the one hand, the law of meaning distribution, that relates the frequency rank of a word with its number of meanings (the most frequent word has rank 1, the 2nd most frequent word has rank 2,. . .). On the other hand, the meaning-frequency law, that relates the frequency of words to their number of meanings.

Both Zipfian laws of meaning indicate the more frequent words tend to have more meanings and their mathematical definition takes the form of a power law [3, 5]. The relationship between the number of meanings of a word, $\mu$, and word frequency, $f$, Zipf's meaning-frequency law, follows approximately

$$\mu \propto f^{\delta}, \tag{1}$$

repository/browse/ELRA-S0407/. The DIEC2 dictionary is available at https://dlc.iec.cat A Zenodo repository has been set up containing: 1. Preprocessed dataset material from both CTILC and DIEC2 corpora (https://zenodo.org/record/4120887/files/DIEC2_CTILC_senseCG.zip) 2. Preprocessed dataset from both Glissando and DIEC2 corpora (https://zenodo.org/record/4120887/files/DIEC2_GLISSANDO_senseCG.zip) 3. FreeLing configuration and execution commands used to process the above datasets (https://zenodo.org/record/5547977/files/freeling.zip).

**Funding:** PRO2020-S03 (RCO03080 Lingüística Quantitativa) from Institut d'Estudis Catalans. (https://www.iec.cat/) PRO2021-S03 HERNANDEZ from Institut d'Estudis Catalans. (https://www.iec.cat/) JB, RFC and AHF are funded by the grant TIN2017-89244-R from Ministerio de Economia, Industria y Competitividad (Gobierno de España) (https://www.cnio.es/) JB, RFC and AHF are supported by the recognition 2017SGR-856 (MACDA) from AGAUR (Generalitat de Catalunya). (https://agaur.gencat.cat/ca/inici) The Institut d'Estudis Catalans (https://www.iec.cat/) provided the following datasets: (1) the normative dictionary of the Catalan language (DIEC2), and (2) the written corpus CTILC.

**Competing interests:** The authors have declared that no competing interests exist.

where $\delta \approx 1/2$ [5–7]. The law of meaning distribution that relates the meanings $\mu$ of a word with its frequency rank $i$ as

$$\mu \propto i^{-\gamma}, \tag{2}$$

where $\gamma \approx 1/2$ [6, 8].

Interestingly, G. K. Zipf never investigated the meaning-frequency law empirically but deduced it from the law of meaning distribution [5, 9] and the popular rank-frequency law, relating $f$ (the frequency of a word) with $i$ (its frequency rank) approximately as [1, 10]

$$f \propto i^{-\alpha}. \tag{3}$$

Zipf deduced Eq 1 with $\delta = 1/2$ from $\alpha = 1$ (for Zipf's rank-frequency law), and $\gamma = 1/2$ (for the law of meaning distribution) [3, 5]. Recently, it has been shown that the three exponents of the power laws are related by [6, 8]

$$\delta = \frac{\gamma}{\alpha}. \tag{4}$$

It should be noted that a *weak* version of Zipf's meaning-frequency law simply indicates that there is a positive correlation between word frequency ($f$) and the number of meanings ($\mu$) what has been connected with a family of Zipfian optimization models of communication [11]. To date, experimental evidence has been accumulating for Zipf's law of meaning distribution, either fitting a power law function or computing the correlation between word frequency and meaning [7, 12–15].

Thus, there is new empirical evidence for the weak version of Zipf's law of meaning distribution on eight languages from different language families (Indo-European, Japonic, Sino-Tibetan and Austronesian), also retrieving exponents of Zipf's law of meaning distribution ($\gamma$) between 0.21 and 0.51 depending on the language (see [12] for details). The results of Bond et al (2019) [12] are consistent with previous works that they review and that have verified the correlation between the frequency of words and their meanings [7, 15] even in child language and language-directed speech [13, 14]. In fact, Bond et al (2019) already noted the influence of both binning size and Zipf's law deviations on the predictive power of the Zipfian laws for the meaning although without considering Eq 4. Eq 4 actually involves assuming the validity of Zipf's rank-frequency law, which has previously been seen not always happen, either due to the appearance of more than one regime in the distribution of words [16–18] or because the data fits better with other mathematical functions [17, 19, 20].

This work is the first empirical study of Zipf's laws of meaning in Catalan and also the first one that considers two sources of different modality for word frequency: an speech corpus and a written one. Our study consists of investigating the exponents of Zipf's meaning-frequency law (Eq 1) and meaning distribution (Eq 2) and then to test the validity of the relationship between the exponents (Eq 4) previously proposed [6, 8]. As far as we know this is the first time that the three Zipfian laws have been analyzed together empirically in one language: Catalan.

## On Catalan

Catalan is a Romance language spoken in the Western Mediterranean by more than ten million people. Catalan is considered a language between the Ibero-Romance (Spanish, Portuguese, Galician) and Gallo-Romance (French, Occitan, Franco-Provençal) languages [21].

Catalan can be considered a language of intermediate complexity from the quantitative perspective of information theory [22]: Catalan has an intermediate level of entropy rate of 5.84,

with languages mean around 5.97±0.91 [23] and morphological complexity (in terms of word complexity, with Catalan ranked in 202 position over 520 languages [24]). As it happens with other Romance languages Catalan does have a great inflectional variability [21], especially in verbs but also in nouns and adjectives [25] with some lexical peculiarities [26]. Besides, derivation is a very productive procedure in the formation of new words in Catalan, suffixation being the most important (above the prefixation and infixation) [27], as it is also usual in other Romance languages [21]. Despite the limited geographical extension of Catalan, there are local variations in suffixation processes that have been reviewed in detail (see [27] and references therein).

Catalan is also a language that has recently been studied in depth under the paradigm of quantitative linguistics [28], recovering the best-known linguistic laws in which meaning does not intervene (as is the case of Zipf's rank-frequency law, Herdan-Heaps' law, the brevity law or the Menzerath-Altmann law) in this speech corpus (Glissando) and in its transcripts [29]. In addition, although the issue is still debated [30], it has also been found the lognormal distribution of words, lately proposed as a new linguistic law [4, 29] but, nevertheless, the statistical patterns of meaning in Catalan has not been addressed until the present study.

Since Zipf's pioneering research, one of the most remarkable discoveries on the rank-frequency law in large multi-author textual corpus is that the power law put forward by Zipf (Eq 3) has to be generalized, on a first approximation, as a double power law of the form [16–18, 31–33],

$$
f \sim \begin{cases} i^{-\alpha_1} & \text{for } i \leq i^* \\ i^{-\alpha_2} & \text{for } i \geq i^*, \end{cases}
\tag{5}
$$

where $\alpha_1$ is the exponent for the low rank (high frequency) power-law regime corresponding to Eq 3, $\alpha_2$ is that of the high rank (low frequency) regime, and $i^*$ is the breakpoint rank. In the British National Corpus it was found that $\alpha_1 = 1$ and $\alpha_2 = 2$ [16]. Precisely, these two scaling regimes in Zipf's rank-frequency law were referred to as the *kernel lexicon*, for the most frequent words usually shared by the majority of speakers of the language, while the rarer words would be part of the so-called *unlimited lexicon*, formed by less common, more specialized or technical words or, in the case of more extensive diachronic corpus, that have fallen into disuse [16, 31].

Here we will provide evidence, in a novel way, that such a double power-law regime also applies to Zipf's laws of meaning. First, the law of meaning distribution becomes

$$
\mu \sim \begin{cases} i^{-\gamma_1} & \text{for } i \leq i^* \\ i^{-\gamma_2} & \text{for } i \geq i^*, \end{cases}
\tag{6}
$$

where $\gamma_1$ is the exponent for high frequency regime, corresponding to $\delta$ in Eq 1, and $\gamma_2$ is the exponent for the low frequency regime. Second, the meaning-frequency law becomes

$$
\mu \sim \begin{cases} f^{\delta_2} & \text{for } f \leq f(i^*) \\ f^{\delta_1} & \text{for } f \geq f(i^*), \end{cases}
\tag{7}
$$

where $f(i^*)$ is the frequency of the word of rank $i^*$, $\delta_2$ is the exponent for the low frequency regime and $\delta_1$ is the exponent for the high frequency regime (Eq 1). Notice that, according to our convention, the subindex 1 (in $\alpha_1$, $\gamma_1$ and $\delta_1$) is used to refer to high frequencies (low ranks) while the subindex 2 (in $\alpha_2$, $\gamma_2$ and $\delta_2$) is used to refer to low frequencies (high ranks).

The remainder of the article is organized as follows. In Section On Materials and methods, we introduce the materials used, i.e. two Catalan corpora, one based on written texts (CTILC)

and the other based on transcribed speech (Glissando), as well as the DIEC2 normative dictionary [34], from which the number of meanings (or polysemy of words) were obtained. We also present the lemmatization process with FreeLing [35] and the binning method used, following Zipf [5]. In Section Result theoretic model selection, we first present an empirical exploration of the three Zipfian laws outlined above assuming a single regime followed by and analysis assuming two regimes. We will show that Eq 4 offers a poor prediction of $\delta$ when a single power-law regime as in Zipf's classic work for the multi-author corpora described above. In contrast, the assumption of two regimes, namely

$$\delta_1 = \frac{\gamma_1}{\alpha_1} \tag{8}$$

$$\delta_2 = \frac{\gamma_2}{\alpha_2}, \tag{9}$$

improves the quality of predictions. Finally, in Section Discussion, we discuss these results for Catalan in the context of previous studies [12–15] and revisit the hypothesis of the existence of a core vocabulary and a peripheral vocabulary [16, 18, 31].

## Materials and methods

### Materials

One of the most important institutional contributions to Catalan corpus linguistics is the *Corpus Textual Informatitzat de la Llengua Catalana* (CTILC), a corpus that covers Catalan written language in texts from 1833 to 1988 (available at https://ctilc.iec.cat/). Based on this corpus, of which an expanded and updated version is in progress, the eminent linguist Joaquim Rafel i Fontanals was able to create a frequency dictionary of Catalan words [36]. By way of example, this implies that a trendy word at present (unfortunately) such as *coronavirus* does not appear in CTILC corpus and, however, there are words that have fallen into disuse from the 19th century such as *coquessa* (a cooker who is hired to make meals on holidays).

This frequency dictionary contains more than 160,000 distinct lemmas, with about 52 million word tokens in total: about 29 million tokens from non-literary texts (56%) and 23 million tokens from literary texts (the remaining 44%) [36]. A more recent corpus is the Glissando corpus recorded in 2010, that contains more than 12 hours of read speech—news—and 12 more hours of studio recordings of dialogues which have been transcribed and aligned to the voice signal [37]. In fact, Glissando is an annotated speech corpus, that is bilingual (Spanish and Catalan) and consisting of the recordings of twenty eight speakers per language, with about 93,000 word tokens and more than 5,000 types in Catalan [29, 37].

### Methods

**Preprocessing.** The empirical study of meaning, from the experimental perspective of quantitative and corpus linguistics, has several methodological challenges that were already pointed out by Zipf in his seminal study [5]. In fact, Zipf devoted the first pages of [5] to the reflection and justification of the corpus he chose in his analysis. Zipf's pioneering work assumed the concept of corpus lemmatization, although without explicitly citing it [3, 5]. Zipf refers to a "dictionary forms"_ and concludes, interestingly, that "...*we have no reason to suppose that any 'law of meanings' would be seriously distorted if we concentrated our attention upon lexical units and simply ignored variations in number, case, or tense.*" (see in [3], p. 29]).

In general, this type of approach usually starts from a *study corpus* (either written or transcribed from orality) that is not lemmatized. Lemmatization is the process of grouping

together the different forms of a word (inflectional forms or derivationally related) in a single linguistic element, identified by the word's *lemma* or *dictionary form*, since it is the form that is usually found in dictionaries [38].

This process of lemmatization is necessary to compare the study corpus with the dictionary corpus in order to establish which subset of words from the study corpus are present in the dictionary. In general, not all the words from a study corpus will be in the dictionary although the dictionary will be responsible for indicating the number of meanings of each word. The situation is depicted in Fig 1, where the corpora (the spoken corpus and written corpus) and the dictionary used in this work are displayed as a Venn diagram.

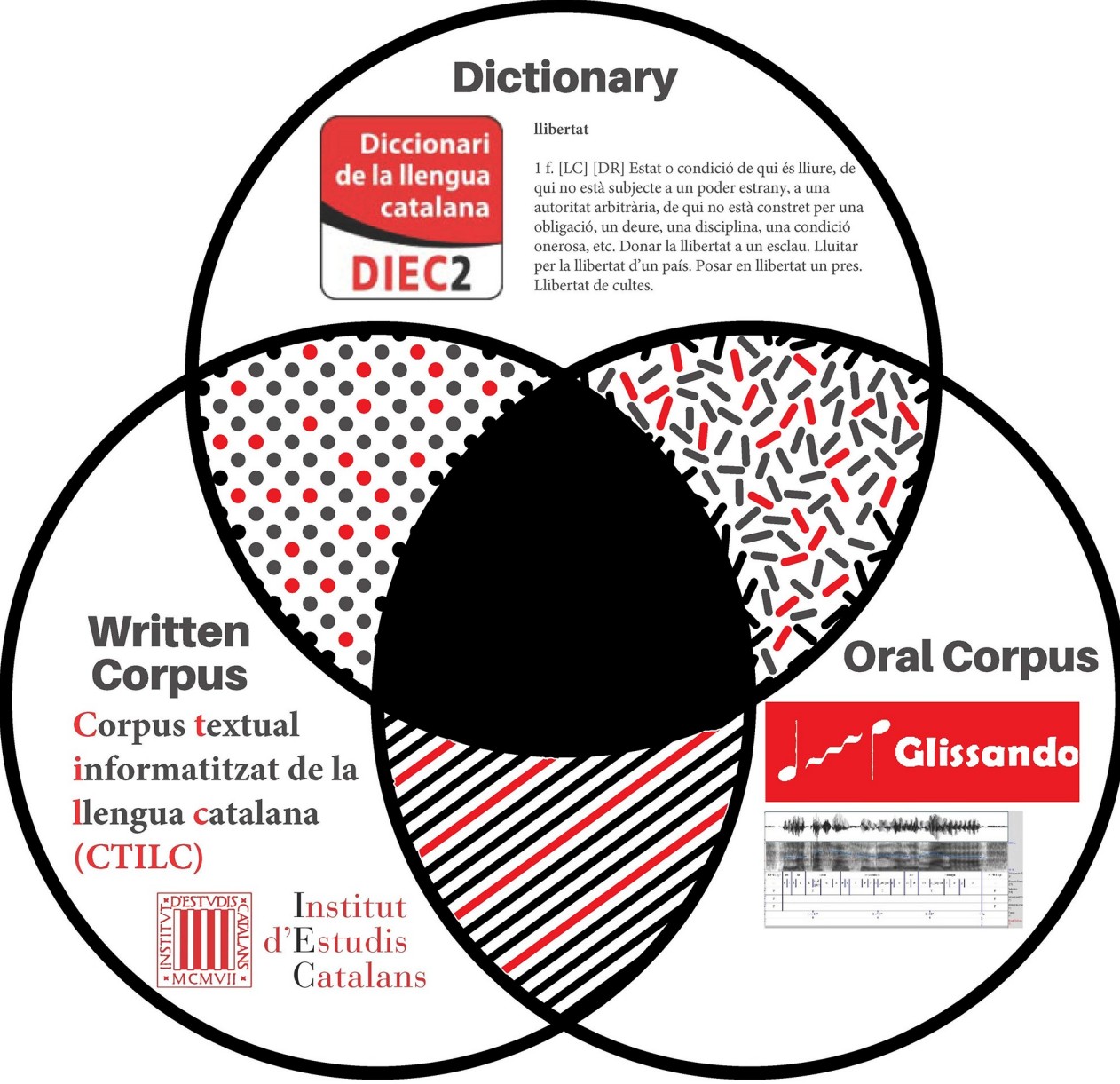

**Fig 1. Graphical representation of the different sources used.** Venn diagram where the sources used in the present study are displayed as a sets: the normative dictionary of the Catalan language (DIEC2), providing the number of meanings of each word; the written corpus CTILC, providing the basis for the descriptive dictionary and the dictionary of word frequencies of Catalan; and the speech corpus Glissando, that is not initially lemmatized, but that has been lemmatized here with FreeLing [35].

**Table 1. Summary of the data.** Number of tokens, different lemmas and availability details of the different sources used in the present study. The lemmas of the Glissando corpus were obtained after lemmatization with Freeling [35]. For the DIEC2 dictionary, and its intersection with the respective corpora, the number of tokens is not applicable because the dictionary only includes lemmas.

| Corpus | Tokens | Lemmas | Available (website link) |
|---|---|---|---|
| CTILC | 52 million | 167,079 | https://ctilc.iec.cat/ |
| Glissando | 93,069 | 4,510 | ELRA Catalogue |
| DIEC2 | — | 70,170 | https://dlc.iec.cat/ |
| CTILC ∩ DIEC2 | — | 52,578 | Zenodo |
| Glissando ∩ DIEC2 | — | 3,083 | Zenodo |

Our analysis of the relationship between the number of meanings of a lemma and its frequency consists in combining information about the number of meanings from the official dictionary of the Catalan language (*Diccionari de la llengua catalana*, DIEC2 [34]) with two sources for the frequency: written language (*Corpus Textual Informatitzat de la Llengua Catalana*, CTILC) and speech (*Glissando Corpus*). As shown in Fig 1, we work with a subset of each corpus, that is defined by the intersection of the study corpus (spoken or written) with the dictionary, which implies necessarily a decrease in the number of lemmas with respect to the initial corpus. Finally, the total number of different lemmas of each corpus is specified in Table 1. The intersection of the CTILC corpus with DIEC2 yields 52, 578 lemmas, while the intersection of the Glissando corpus with DIEC2 yields 3, 083 lemmas.

CTILC is manually lemmatized and annotated with Parts-of-Speech (PoS) tags and Glissando contains the direct transcriptions of spoken dialogues. In order to be able to perform the same analysis in both corpora, we resorted to FreeLing [35] to lemmatize the Glissando corpus. FreeLing is an open-source library offering a variety of linguistic analysis functionalities for more than 12 languages, including Catalan. Details can be found at http://nlp.cs.upc.edu/freeling.

More specifically, the natural language processing layers used in this work were:

**Tokenization & sentence splitting:** Given a text, split the basic lexical terms (words, punctuation signs, numbers, etc.), and group these tokens into sentences.

**Morphological analysis:** Find out all possible Parts-of-Speech (PoS) for each token.

**PoS-Tagging and Lemmatization:** Determine the right PoS for each word in its context. Determining the right PoS allows inferring the right lemma in almost all cases.

**Named Entity Recognition:** Detect proper nouns in the text, which may be formed by one or more tokens. We used only pattern-matching based detection relying mainly on capitalization.

We used FreeLing to perform PoS-tagging, lemmatization, and proper noun detection on Glissando corpus. We then filtered out all tokens marked as punctuation, number, or proper noun, before proceeding to count occurrences of each lemma and cross them with DIEC2. FreeLing was configured deactivating date, number, and multiword detection. In addition, proper noun configuration was changed from the default (gluing toghether proper nouns as multiwords) to keep proper noun tokens separated. Finally dictionary entries for contractions and related retokenization options were set to avoid contraction splitting. The link to the configuration file and execution command used to process the files is in Data availability section.

Of the 93, 069 tokens remaining after the filtering, 82, 838 (89%) were lemmatized by FreeLing, producing 4, 510 different lemmas. The remaining 11% tokens correspond to

interjections, hesitations, half-words, foreign words (Spanish or English), colloquial expressions, non-capitalized proper nouns, or transcription errors (where the transcription was made phonetically, and not with the right form of the intended word) and were left out of the study.

**Fitting method.** After intersecting each corpus with DIEC2 (Fig 1), we analized the resulting data sets and fitted the power law functions of the three Zipfian laws, using linear Least Squares (LS) on a logarithmic transoformation of both axes: the rank-frequency law, the law of meaning distribution and the meaning-frequency law, as defined in Eqs 1–3, respectively.

Following Baayen's method [39], we calculated the most likely breakpoint, $i^*$ (the data point where the two regimes cross) in both corpora. The method consists of scanning all possible breakpoints and compute, for each, the deviation (sum of squared errors) or deviance, as Baayen's refers to it, between the real points and Eq 5. This breakpoint is the rank that minimizes the deviance. Some care is required in this method because there might be more than one local optimum, as we will see.

Since we are studying three interrelated Zipfian laws (rank-frequency law, law of meaning distribution and meaning-frequency law), we decided, for simplicity, to transfer the breakpoint obtained for the rank-frequency law to the other double regime laws. The translation of the breakpoint to the double regime law of meaning distribution (Eq 6) is immediate: the breakpoint, $i^*$, is the same as in the rank-frequency law. The translation of the $i^*$ to the breakpoint of the double regime meaning-frequency law (Eq 7) requires computing $f(i^*)$. The value of $f(i^*)$ is calculated from the mean of the frequencies of the data in the bin where $i^*$ is located. $i^*$ and $f(i^*)$ are used to complete the fitting and retrieve the exponents of the double regime laws. Accordingly, for the law of meaning distribution, we fit Eq 6, where the relevant parameters are $\gamma_1$ and $\gamma_2$. For the meaning frequency law, we fit Eq 7 where the relevant parameters are $\delta_1$ and $\delta_2$.

**Curve smoothing.** The fitting method described above is applied to two kinds of data: the raw data and the smoothed data. In the raw data approach, every point of the curve corresponds to a "word". No bucketing or binning was applied. The smoothed data approach follows from Zipf, who applied a linear binning technique to reduce noise in his analysis of the law of meaning distribution [3, 5]. In previous work [13], Zipf's analysis of the law in English [3] was revisited and values for exponents very close to those already obtained by Zipf were retrieved but, surprisingly, with sources of data differing from Zipf's work: in the case of the law of meaning distribution, $\gamma$ achieving a value of 0.5 as in Zipf's pioneering research [5].

Given the robustness of this method, the data smoothing approach consisted of estimating the exponents of the three Zipfian laws after applying a linear binning on the data sets. In particular, we applied equal-size binning ($K$ bins, each with $n/K$ data points), in the sense of having the same number of data points in each bin, considering $n$ the total number of lemmas of each subcorpus studied (Table 1). In equal-size binning, resulting bins have an equal number of observations in each group. Bin sizes have been chosen from divisors of the number of data points (lemmas) in each corpus to warrant that every bin has the same number of points and that no data point is lost. CTILC ∩ DIEC2 contains 53, 578 lemmas and Glissando ∩ DIEC2 contains 3, 083 lemmas.

## Information theoretic model selection

We applied information theoretic model selection [40] to assess if a two-regime model approach provides a better description of the data than a single-regime model. In particular, we employed the corrected Akaike Information Criterion (AICc) and the Bayesian Information Criterion (BIC) [41]. Both scores favour models that are more likely but penalizing for

their complexity, measured with $k$, the number of parameters. Both scores take into consideration $n$, the number of observations.

AICc is a variant of the plain AIC that incorporates a correction for $n$. BIC and the plain AIC are defined as

$$\text{AIC} = 2k - 2\,\ln(\hat{L})$$

$$\text{BIC} = k\,\ln(n) - 2\,\ln(\hat{L}),$$

where $\hat{L}$ is the likelihood of the model. In turn, AICc is defined as [40]

$$\text{AICc} = \text{AIC} + \frac{2k^2 + 2k}{n - k - 1} \tag{10}$$

AIC and BIC values were computed using the R library `stats`. AICc was computed applying the values delivered by R to Eq 10.

For each Zipfian law, $k = 3$ for the one-regime models (intercept, slope and error variance). The number of parameters of the two-regime models varies. For Zipf's rank-frequency law, $k = 5$ (intercept, 2 slopes, breakpoint and error variance). For Zipf's law of meaning distribution and Zipf's meaning-frequency law, $k = 4$ (intercept, 2 slopes and error variance). Notice that the breakpoint here is not a free parameter because it is set by Zipf's rank-frequency law.

## Results

### One regime analysis

Figs 2 and 3 show Zipf's rank-frequency law for the CTILC corpus and the Glissando corpus, respectively. Similarly, Figs 4 and 5 show the meaning distribution law for CTILC and Glissando, respectively, and Figs 6 and 7 show the meaning-frequency law for CTILC and Glissando, respectively. The values of the exponents of each law are summarized in Table 2, both in the case of equal-size binning and also when no smoothing is performed. Interestingly, $\gamma$ approaches 0.5 in CTILC as the bin size increases and the difference between $\delta$ and $\delta'$ (the predicted value of $\delta$ from $\alpha$ and $\gamma$ in Eq 4), is slightly reduced when binning is used in both corpora.

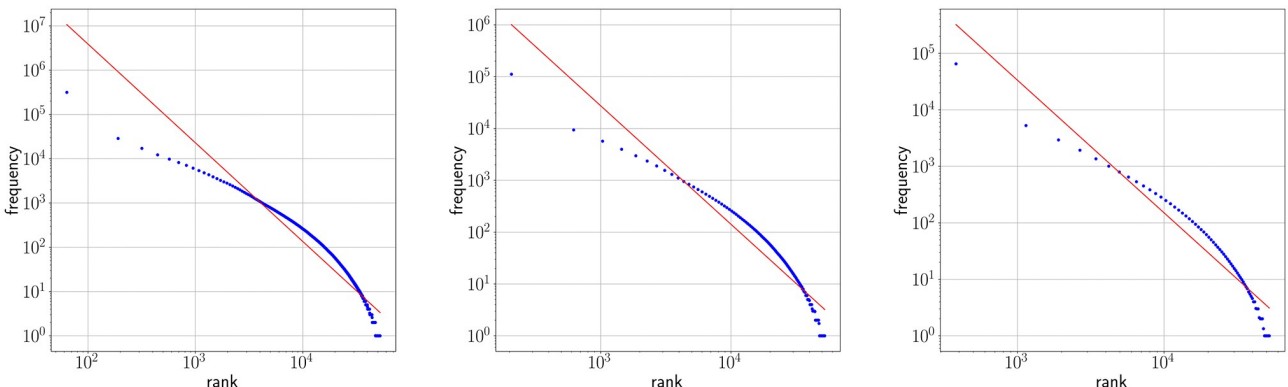

**Fig 2. Zipf's rank-frequency law in CTILC corpus.** Average frequency ($f$) as a function of rank ($i$) after applying equal-size binning (blue). The best fit of a power law is also shown (red). Left: bin size of 127 words. Center: bins size of 414 words. Right: bin size of 762 words.

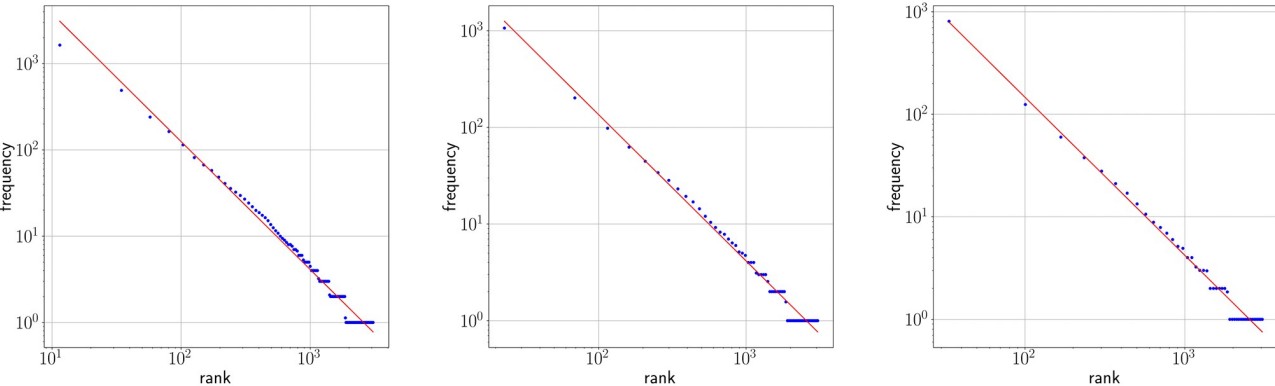

**Fig 3. Zipf's rank-frequency law in Glissando corpus.** Average frequency ($f$) as a function of rank ($i$) after applying equal-size binning (blue). The best fit of a power law is also shown (red). Left: bin size of 23 words. Center: bins size of 46 words. Right: bin size of 67 words.

## Two regime analysis

After visual inspection of the rank-frequency law (Figs 2 and 3), the appearance of at least two regimes, i.e. straight lines in log-log scale with different slopes, is suggested in the CTILC written corpus (Fig 2). That is a well-known feature arising in large multi-author corpora [16–18, 31–33]. However, interestingly, these two regimes are apparently not visually observed in the Glissando speech corpus (Fig 3) but they might be hidden in the greater dispersion of the points in meaning distribution, despite the linear binning, that can be seen in the speech corpus (Fig 5) with respect to CTILC (Fig 4).

To confirm the presence of two regimes in the CTILC corpus and shed light on the possible presence of two regimes in the Glissando corpus, we perform a careful analysis using Baayen's method of breakpoint detection [39]. As can be seen in Fig 8, the deviance has only a single minimum in the case of CTILC corpus, which allows to determine a breakpoint easily. Besides, a global minimum and a and and additional local minimum are found in the Glissando corpus (Fig 9): in the first binning (23 words per bin), the global minimum does correspond to a meaningful two-regime breakpoint but, in the other two binnings (46 and 67 words per bin), the global mininum corresponds to a spurious result due to the abundance of hapax legomena

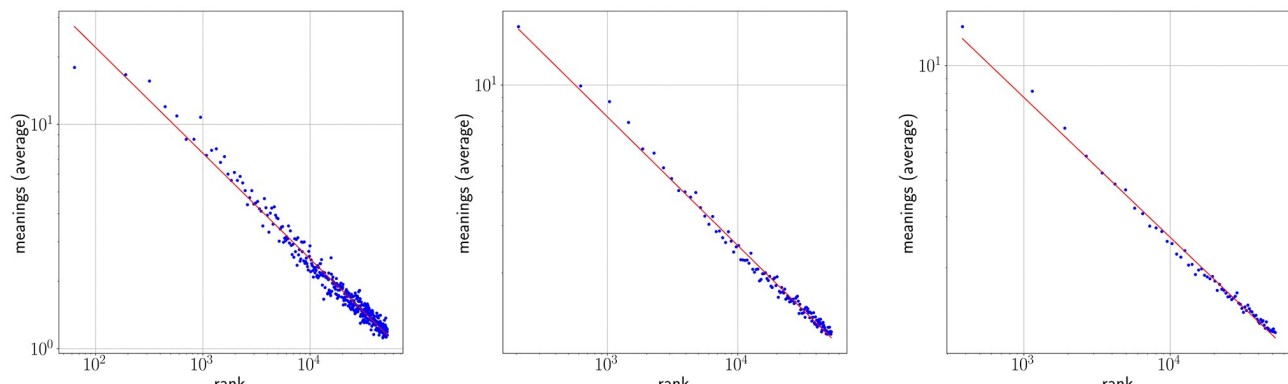

**Fig 4. Zipf's law of meaning distribution in CTILC corpus.** Average number of meanings ($\mu$) as a function of frequency rank ($i$) after applying equal-size binning (blue). The best fit of a power law is also shown (red). Left: bin size of 127 words. Center: bin size of 414 words. Right: bin size of 762 words. Sources: Catalan words in CTILC, using DIEC2 meanings and CTILC frequencies.

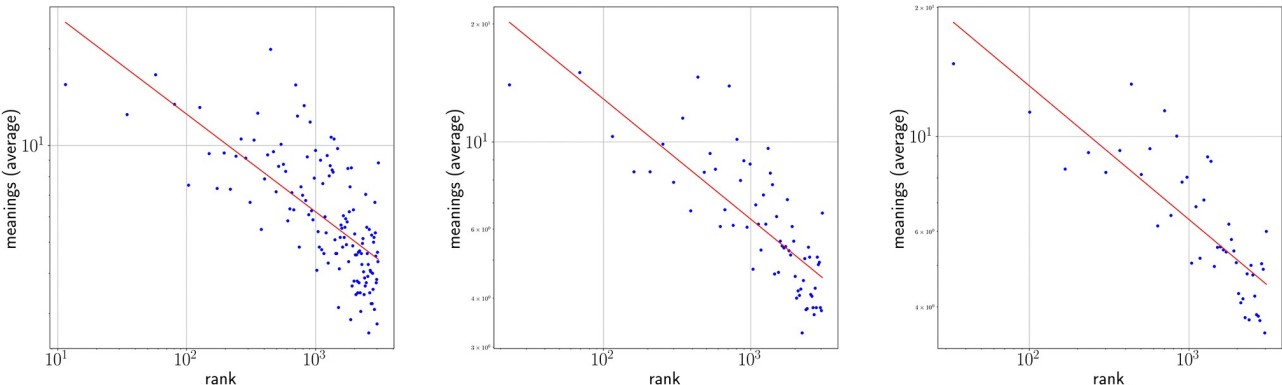

**Fig 5. Zipf's law of meaning distribution in Glissando corpus.** Average number of meanings ($\mu$) as a function of frequency rank ($i$) after applying equal-size binning (blue). The best fit of a power law is also shown (red). Left: bin size of 23 words. Center: bin-size of 46 words. Right: bin-size of 67 words. Sources: Catalan words in Glissando, using DIEC2 meanings and Glissando frequencies.

in the tail of the rank distribution, a phenomenon that can also be observed in the first binning (Fig 9). This spurious minimum in deviance can be explained by an artifact of the deviance minimization procedure for breakpoint detection, that becomes too sensitive to the concentration of points in the tail and the scarcity of points in other parts of the curve when bin size increases [39]. As a result it is observed that when bin size increases, the local minima in deviance decreases (see right panels in Fig 9).

The value of the breakpoint increases with the size of the bin. This value is approximately 20, 606 (127 words per bin), 22, 149 (414 words per bin) and 23, 241 (762 words per bin) in the CTILC corpus (Fig 8). In the Glissando corpus (Fig 9), the non-spurious breakpoint is found at 333, 5 (23 words per bin), 437, 0 (46 words per bin) and 502, 5 (67 words per bin). Evidently, the size of the corpus influences the the values of these breakpoints.

Fig 10 shows the meaning-frequency law with this two regime analysis in CTILC and Glissando corpora, and Fig 11 shows the two-regime in law of meaning distribution. In both cases, by varying the bin size, the fitting curve for the CTILC corpus has a clear breakpoint that divides the two regimes, while for the Glissando corpus this breakpoint is not so clear visually,

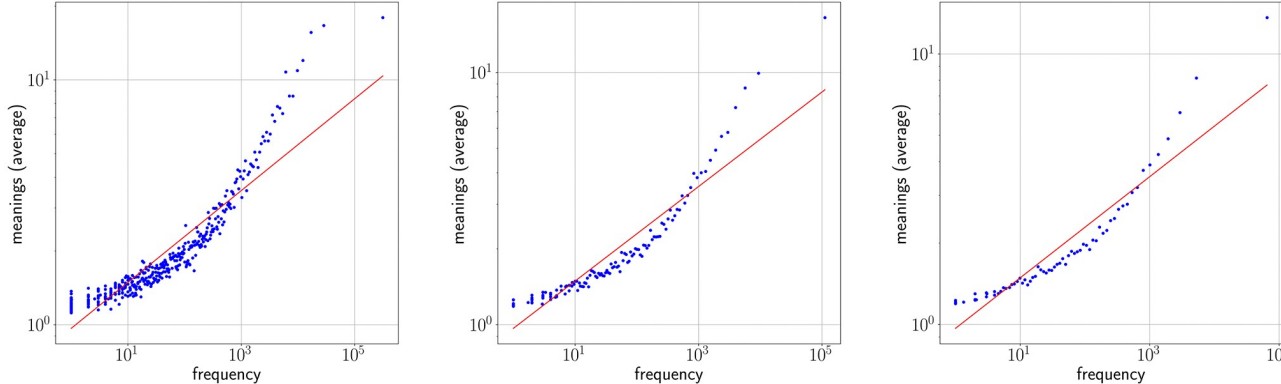

**Fig 6. Zipf's meaning-frequency law in CTILC corpus.** Average number of meanings ($\mu$) as a function of frequency ($f$) after applying equal-size binning (blue). The best fit of a power law is also shown (red). Left: bin size of 127 words. Center: bin size of 414 words. Right: bin size of 762 words. Sources: Catalan words in CTILC, using DIEC2 meanings and CTILC frequencies.

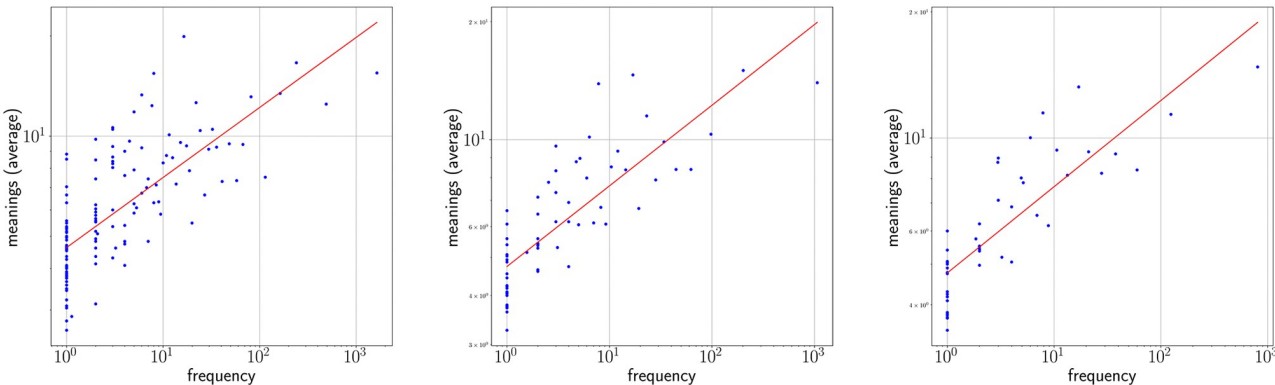

**Fig 7. Zipf's meaning-frequency law in Glissando corpus.** Average number of meanings ($\mu$) as a function of frequency ($f$) after applying equal-size binning (blue). The best fit of a power law is also shown (red). Left: bin size of 23 words. Center: bin-size of 46 words. Right: bin-size of 67 words. Sources: Catalan words in Glissando, using DIEC2 meanings and Glissando frequencies.

as there is greater dispersion of data in the speech corpus and, consequently, a weaker correlation.

Table 3 summarizes the exponents of the Zipfian laws obtained in each of the two regimes for the CTILC and Glissando corpora. The difference between $\delta_1$ and its predicted value $\delta'_1$ from Eq 8, and the difference between $\delta_2$ and its predicted value $\delta'_2$ from Eq 9, are, in general, smaller than when a single regime was assumed in Table 2, providing support for the reality of two-regime structure.

## Model selection

Now we turn to the question of which of the two approaches, a two-regime model or a one-regime model, actually provides a better description of the data. To address the problem of the trade-off between the goodness of fit of the model and its number of parameters, we recur to information theoretic model selection [40].

Tables 4, 5 and 6 show AICc and BIC values for each Zipfian law, corpus and kind of binning. All values favour two-regime models over one-regime models, except for the law of meaning distribution and the meaning-frequency law on the Glissando corpus with no binning, where the scores for each model are very close to each other.

**Table 2. One regime analysis (CTILC and Glissando corpora).** The exponents of the Zipfian laws: the rank-frequency law ($\alpha$, Eq 3), the law of meaning distribution ($\gamma$, Eq 2) and meaning-frequency law ($\delta$, Eq 1). $\delta'$ is the exponent $\delta$ predicted by Eq 4, obtained from $\alpha$ and $\gamma$. For estimating the values of the parameters, we have used Least Squares (LS) on a logarithmic transformation of both axes. Concerning equal-size binning, see the Methods section for the rationale behind the choice of the bin sizes.

| Binning | Corpus | bin size | $\alpha$ | $\gamma$ | $\delta$ | $\delta'$ |
|---|---|---|---|---|---|---|
| No binning | CTILC ∩ DIEC2 | - | 2.199 | 0.388 | 0.154 | 0.176 |
| | Glissando ∩ DIEC2 | - | 1.459 | 0.261 | 0.184 | 0.178 |
| Equal-size | CTILC ∩ DIEC2 | 127 | 2.228 | 0.471 | 0.187 | 0.211 |
| | | 414 | 2.286 | 0.478 | 0.187 | 0.209 |
| | | 762 | 2.347 | 0.484 | 0.187 | 0.206 |
| | Glissando ∩ DIEC2 | 23 | 1.483 | 0.304 | 0.210 | 0.205 |
| | | 46 | 1.513 | 0.306 | 0.206 | 0.202 |
| | | 67 | 1.542 | 0.312 | 0.205 | 0.202 |

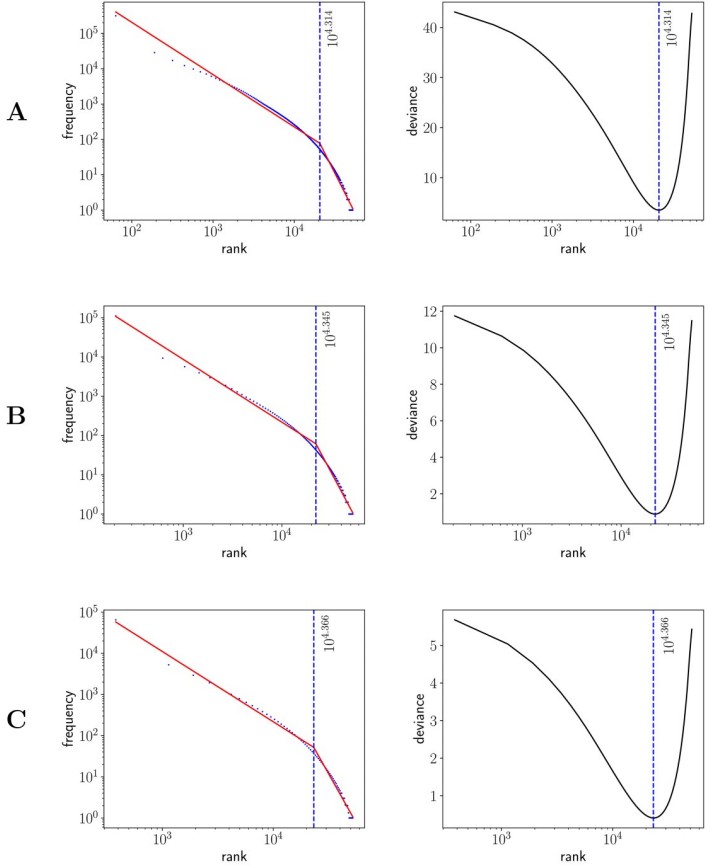

**Fig 8. Breakpoint analysis for Zipf's rank-frequency law in CTILC corpus.** Breakpoints ($i^*$) are depicted as blue dashed lines. Left: frequency ($f$) as a function of rank ($i$). The best fit of a two-regime power law is also shown (red). Right: Baayen's deviance as a function of rank ($i$). The choice of the bin sizes is the same as in Table 3 and influences the breakpoint. Row A) 127 words per bin and breakpoint 20, 606, row B) 414 words per bin and breakpoint 22, 149, and row C) 762 words per bin and breakpoint 23, 241.

## Discussion

### Zipf's laws of meaning and robustness of the equation between exponents

Zipf's laws of meaning have been verified in Catalan. Assuming a single regime and applying equal-size binning to the CTILC corpus, the $\gamma$ exponent of the law of meaning distribution that Zipf found for the word meaning distribution law in English [3, 5] was recovered approximately, $\gamma \approx 1/2$, (Table 2), consistently with previous work [6, 8]. This is not the case of the Glissando speech corpus where $\gamma \approx 0.304 - 0.312$. Nonetheless, considering the results for the fitting of a single slope ($\gamma$) for the law of meaning distribution, the $\gamma$ values obtained here are consistent with previous works [13, 15] that extended the fitting of this law to non-Indo-European languages [12]. As shown in previous research, verified here again for Catalan, there is a certain variability in the exponents of the Zipfian laws according to the size of the binning [12, 13, 15].

However, there were clear deviations in Zipf's rank-frequency law exponent, i.e. $\alpha$, with respect to Catalan ($\alpha = 1.42$ in [29] using different methods) and other languages in normal conditions [42, 43]. Consequently, this fact affected the exponent $\delta'$ obtained indirectly with Eq 4 (see Table 2). These considerations notwithstanding, $\delta'$ gives approximately a similar

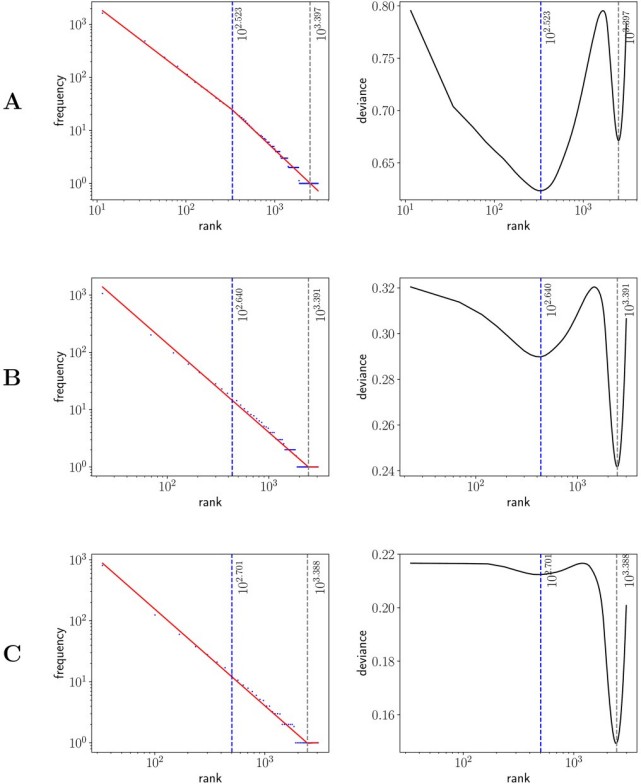

**Fig 9. Breakpoint analysis for Zipf's rank-frequency law in Glissando corpus.** Blue dashed lines and gray dashed lines are used to indicate, respectively, the first and the second local minimum of deviance. The 1st local minimum is taken as the meaningful breakpoint ($i^*$). Left: frequency ($f$) as a function of rank ($i$). The best fit of a two-regime power law is also shown (red). Right: Baayen's deviance as a function of rank ($i$). The choice of the bin sizes is the same as in Table 3 and influences the first non-spurious breakpoint. Row A) 23 words per bin and breakpoint 333.5, row B) 46 words per bin and breakpoint 437.0, and row C) 67 words per bin and breakpoint 502.5.

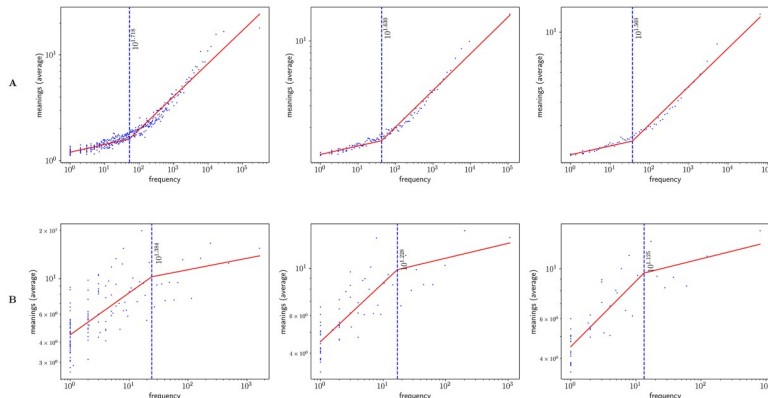

**Fig 10. The meaning-frequency law in the CTILC and Glissando corpora.** Row A) CTILC corpus and row B) Glissando corpus. The choice of the bin sizes is the same as in Table 3. Breakpoints ($i^*$) are depicted as dashed blue lines.

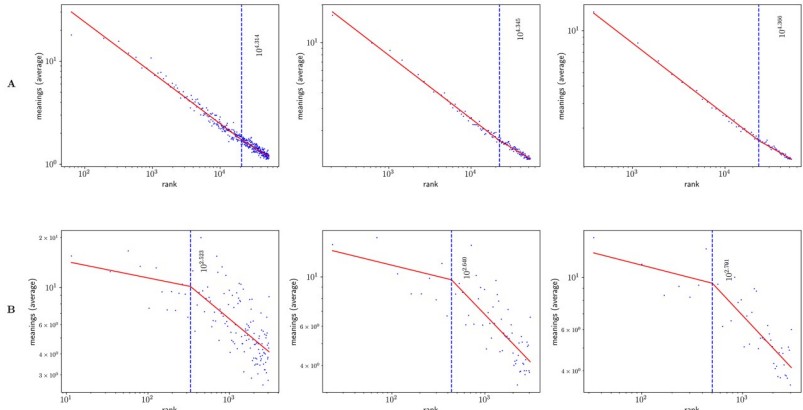

**Fig 11. The law of meaning distribution in the CTILC and Glissando corpora.** Row A) CTILC corpus and row B) Glissando corpus. The choice of the bin sizes is the same as in Table 3. Breakpoints ($i^*$) are depicted as dashed blue lines.

**Table 3. Two regime analysis (CTILC and Glissando corpora).** The exponents of each regime of the Zipfian laws: the rank-frequency law ($\alpha_1$ and $\alpha_2$), the law of meaning distribution ($\gamma_1$ and $\gamma_2$) and meaning-frequency law ($\delta_1$ and $\delta_2$). $\delta'_1$ is the exponent $\delta_1$ predicted by Eq 9 while $\delta'_2$ is the exponent $\delta_2$ predicted by Eq 8. To obtain the exponents, we used LS following Baayen's method [39]. Concerning equal-size binning, see the Methods section for the rationale behind the choice of the bin sizes. Subindexes of the exponents correspond to the regimes according to Eqs 5–7.

| CTILC | | | | | | | | | |
|---|---|---|---|---|---|---|---|---|---|
| Binning | bin size | $\alpha_1$ | $\alpha_2$ | $\gamma_1$ | $\gamma_2$ | $\delta_1$ | $\delta'_1$ | $\delta_2$ | $\delta'_2$ |
| No binning | – | 1.414 | 4.483 | 0.419 | 0.298 | 0.275 | 0.296 | 0.052 | 0.066 |
| Equal-size | 127 | 1.480 | 4.560 | 0.494 | 0.402 | 0.312 | 0.334 | 0.073 | 0.088 |
| | 414 | 1.600 | 4.707 | 0.503 | 0.388 | 0.297 | 0.314 | 0.068 | 0.082 |
| | 762 | 1.707 | 4.800 | 0.512 | 0.375 | 0.286 | 0.300 | 0.065 | 0.078 |
| **Glissando** | | | | | | | | | |
| No binning | – | 0.853 | 1.537 | 0.065 | 0.287 | 0.059 | 0.076 | 0.194 | 0.187 |
| Equal-size | 23 | 1.281 | 1.583 | 0.098 | 0.405 | 0.071 | 0.076 | 0.262 | 0.256 |
| | 46 | 1.404 | 1.583 | 0.102 | 0.436 | 0.069 | 0.073 | 0.275 | 0.275 |
| | 67 | 1.495 | 1.577 | 0.110 | 0.459 | 0.072 | 0.074 | 0.291 | 0.291 |

**Table 4. AICc and BIC values for the one-regime model (1r) and the two-regime model (2r) for Zipf's rank-frequency law.** $n$ is the number of points.

| CTILC | | | | | | |
|---|---|---|---|---|---|---|
| Binning | bin size | $n$ | AICc | | BIC | |
| | | | 1r | 2r | 1r | 2r |
| No binning | – | 52,578 | 33605.77 | -95008.19 | 33632.38 | -94963.84 |
| Equal-size | 127 | 414 | 244.237 | -788.290 | 256.256 | -768.308 |
| | 414 | 127 | 64.242 | -259.132 | 72.580 | -245.406 |
| | 762 | 69 | 29.878 | -147.498 | 36.211 | -137.280 |
| **Glissando** | | | | | | |
| No binning | – | 3,083 | -6171.082 | -7426.773 | -6152.989 | -7396.605 |
| Equal-size | 23 | 134 | -300.531 | -328.897 | -292.022 | -314.877 |
| | 46 | 67 | -161.454 | -175.757 | -155.220 | -165.717 |
| | 67 | 46 | -109.358 | -121.547 | -104.444 | -113.904 |

**Table 5. AICc and BIC values for the one-regime model (1r) and the two-regime model (2r) for Zipf's law of meaning distribution.** *n* is the number of points.

**CTILC**

| Binning | bin size | n | AICc | | BIC | |
|---|---|---|---|---|---|---|
| | | | 1r | 2r | 1r | 2r |
| No binning | – | 52,578 | -7021.764 | -7185.055 | -6995.154 | -7149.576 |
| Equal-size | 127 | 414 | -1639.696 | -1670.549 | -1627.677 | -1654.543 |
| | 414 | 127 | -620.296 | -659.527 | -611.958 | -648.479 |
| | 762 | 69 | -352.745 | -393.797 | -346.412 | -385.486 |

**Glissando**

| Binning | bin size | n | AICc | | BIC | |
|---|---|---|---|---|---|---|
| No binning | – | 3,083 | 3234.723 | 3228.888 | 3252.816 | 3253.01 |
| Equal-size | 23 | 134 | -155.601 | -163.972 | -147.093 | -152.691 |
| | 46 | 67 | -111.902 | -121.156 | -105.669 | -112.982 |
| | 67 | 46 | -83.455 | -91.648 | -78.541 | -85.309 |

value as the $\delta$ obtained directly from the meaning-frequency law ($\delta' = 0.187 - 0.21$ (CTILC corpus) and $\delta' = 0.20 - 0.21$ (Glissando corpus)) but deviates in both cases from the previously estimated for English [5, 6]. The well-known deviations in Zipf's rank-frequency law (Eq 3) imply that it cannot be assumed that $\alpha = 1$ in general [42, 43], therefore, according to Eq 4, it cannot be expected that $\gamma = \delta$ as in Zipf's early work [5, 12].

Nevertheless, the Eq 4 to obtain $\delta'$ indirectly is even robust in the case of the analysis of one regime without binning in both corpus (Table 2). This is especially interesting given that, in this case, both in the CTILC corpus ($\alpha \approx 2.20$, $\gamma \approx 0.39$) and in Glissando ($\alpha \approx 1.46$, $\gamma \approx 0.26$) the Zipfian exponents are far from the usual ones but $\delta \approx \delta'$. On the other hand, the minimal differences found in $\alpha$ with respect to the previous study of the Glissando corpus [29] (where $\alpha \approx 1.42$) may be due to the fact that here we have worked with a subcorpus of Glissando (Glissando ∩ DIEC2) and with different methods.

On the other hand, after verifying the existence of two regimes in Zipf's rank-frequency law in CTILC corpus (Figs 2 and 8) and in Glissando corpus (Figs 3 and 9), as previous works pointed out in big corpora [16, 18, 31], we have seen that this affects the meaning-frequency law (Fig 10). Then, the analysis of the two regimes in the CTILC corpus allowed us to obtain in the first regime $\gamma_1 \approx 1/2$ and $\delta_1 \approx 0.30$ (with $\delta_1 \approx \delta_1'$), and in the second regime $\gamma_2 = 0.37 - 0.40$ and $\delta_2 \approx 0.06 - 0.07$ (and again $\delta_2' \approx 0.08$). In the case of the Glissando corpus in the

**Table 6. AICc and BIC values for the one-regime model (1r) and the two-regime model (2r) for Zipf's meaning-frequency law.** *n* is the number of points.

**CTILC**

| Binning | bin size | n | AICc | | BIC | |
|---|---|---|---|---|---|---|
| | | | 1r | 2r | 1r | 2r |
| No binning | – | 52,578 | -2826.499 | -6809.194 | -2799.889 | -6773.715 |
| Equal-size | 127 | 414 | -913.675 | -1482.761 | -901.656 | -1466.755 |
| | 414 | 127 | -297.383 | -521.017 | -289.045 | -509.969 |
| | 762 | 69 | 164.332 | -290.327 | -157.999 | -283.390 |

**Glissando**

| Binning | bin size | n | AICc | | BIC | |
|---|---|---|---|---|---|---|
| No binning | – | 3,083 | 3216.13 | 3214.279 | 3234.223 | 3238.4 |
| Equal-size | 23 | 134 | -164.97 | -170.941 | -156.461 | -159.659 |
| | 46 | 67 | -117.389 | -125.200 | -111.155 | -117.027 |
| | 67 | 46 | -87.953 | -96.436 | -83.038 | -90.097 |

first regime (binning size 23) $\gamma_1 \approx 0.1$ and $\delta_1 \approx 0.07$ (with $\delta_1 \approx \delta'_1$), and in the second regime $\gamma_2 = 0.405$ and $\delta_2 \approx 0.26$, and again $\delta_2 \approx \delta'_2$. Results are similar for other bin sizes, as can be seen in Table 3.

Finally, our experimental results show that the relationship (Eq 4) between the three Zipfian exponents [6, 8] is specially robust when we two regimes are assumed as expected from the higher precision in the estimation of the exponents. Again, $\delta$ and $\delta'$ deviate from previously reported for English, $\delta \approx 0.5$ [5, 6, 8]. $\delta'$ may deviate from the expected value because of the value of $\alpha$ retrieved here for Catalan. Besides, $\delta$ may deviate from the expected value because of the two regime structure of the data. Therefore, as a hypothesis to corroborate in future research employing more languages, the variations in $\gamma$ or $\delta$ could be explained by deviations in Zipf's rank-frequency law or the underlying two regime structure reported here.

Our comparison of the error of the theoretical predictions of Eq 4 for the two-regime model against those of the single-regime model, can be seen as a form of model selection based on the mathematical theory of Zipfian power laws [6, 8]. As explained above, that model selection approach has provided indirect support for the existence of these two regimes and, in particular, support for the double power-law model over the single power-law model for each Zipfian law. Alternatively, we have obtained direct support for the two regimes following a formal approach using information theoretic model selection. We have found that the double power-law model fits the data better than the single power-law model in terms of the trade-off between parsimony and goodness of fit (Tables 4–6).

## Two regimes in Zipfian laws: Core vocabulary?

In sum, although we have verified Zipf's meaning-frequency law (Eq 1) between the number of meanings and the frequency of words in Catalan employing Zipf's binning technique [5], data is better-described when two regimes are assumed. These two scaling regimes could be explained simply as the outcome of aggregating texts: previous work indicate that scaling breaks in rank-frequency distributions are a consequence of the mixing and composition of texts and corpora [18].

Previous works showed that some variability was found in the breakpoints in the case of corpus of English, Spanish and Portuguese, depending on the size of the corpus [18]. Here, for both corpora, we have seen that the breakpoint tends to increase with the size of the corpus and also with the size of the bin, and in the case of CTILC corpus (167, 079 lemmas) the breakpoint varies from 20, 606 to 23, 241, and in Glissando corpus (4510 lemmas) from 333.5 to 502.5. Therefore, in the case of Catalan we have also seen this dependence with the size of the corpus.

In our opinion, as seen, the effects of corpus size, composition and heterogeneity previously suggested [17, 18, 42, 44] are not incompatible with the existence of a core and a peripheral vocabulary or "unlimited lexicon" [16, 33], but this dichotomy is not necessarily be an observable property of Zipfian distributions by means of the breakpoint as [18] pointed out. Besides, if we understand the core vocabulary as the real basic vocabulary of a linguistic community at a given time, then Glissando turns out to be a better source than CTILC to capture that subset of the vocabulary, because CTILC corpus mixed sources from different time periods.

One the one hand, the CTILC written corpus includes from literary and journalistic texts to scientific ones, in a time interval of more than a century, with the diachronic variations that this implies. The sum of the size effect and the greater linguistic variability of combining heterogeneous texts in the CTILC corpus could explain the appearance of two regimes in Zipf's rank-frequency law [17, 18, 32] and, as we have seen, consequently, of two regimes in the Zipf's meaning-frequency law. On the other hand, Glissando is a synchronous speech corpus

in which the interlocutors cooperate and circumscribe themselves to a single communicative context, as is often promoted in the systematic *design* of speech corpora [37]. That is, to the usual reduction in the use of rare words that occurs in oral communication, in a pragmatic context with broad Gricean implications (see [45, 46] for a review), it must be added that in the construction of speech corpus like Glissando, literally communicative scenarios are 'designed' [37]. This fact causes that we are not really facing a spontaneous speech corpus, causing a tendency to unify the vocabulary used by the different informants involved in or, at least, reducing the use of infrequent words. Infrequent words are typical of the peripheral vocabulary of multi-author corpus that deal with diverse topics. However, in the speech corpus a greater dispersion of the points in the meaning-frequency distribution is appreciated, but the two regimes are still found.

Lexical diversity is defined as the variety of vocabulary deployed in a text or transcript by either a speaker or a writer [47]. In speech one expects to find a smaller lexical diversity and core vocabulary given the stronger cognitive constraints of spontaneous oral language (Glissando) with respect to written language (CTILC), to which the effects of the lemmatization are added here (see next subsection). Size effects have been shown to influence lexical diversity, even in small corpus [48].

Therefore, other speech corpora of different size and spontaneous speech should be analysed in the future to corroborate these two regimes observed, exploring the size limits that had previously been considered for the appearance of a kernel vocabulary [16]. It also remains as future work to verify if these effects are present in the comparison of spoken and written corpora of other languages and, in addition, if these two regimes appear in other languages, analogously to Catalan, in multi-author texts or whether they appear just as a consequence of text mixing [18, 31], given the influence of semantics on Zipfian distributions [11, 49].

## Lemmatization and binning effects

Glissando is a corpus smaller in size than CTILC and with less thematic variability. However, the effect of binning seems to be added to the effect of corpus size. As Table 1 shows, only the lemmas for which we had their meanings have been analyzed (intersection of each corpus with the DIEC2 dictionary). Thus, there is eventually an order of magnitude difference between the number of lemmas in both corpora (the intersection of CTILC corpus with DIEC2 has 52, 578 lemmas and the intersection of Glissando corpus with DIEC2 only 3, 083 lemmas). The quantitative effects of corpus size have been related to variations of Zipf's law and other linguistic laws in such a way that a larger size in a corpus implies increasing the probability of rare words, so that word frequency distributions are Large Number of Rare Events (LNRE) as [44] explain clearly. Thus, in the case of the speech corpus such as Glissando, smaller in size, also there is a smaller number of rare words, since fewer technical words and jargon are used in orality than in written corpus [50], but the two regimes are nevertheless observed. The lemmatization process implies a decrease in the LNRE by including under the same lemma inflected words, of special importance in Catalan [36], as in other Romance languages [21, 26].

Regarding the deviations observed in the exponents of Zipf's rank-frequency law (Eq 3) also found in other languages [51], notice that by lemmatizing the corpus the morphological complexity and, by extension, the diversity of the vocabulary, is reduced. As explained in the Introduction, Catalan is a Romance language with a rich inflection and derivation [21, 25, 27], that, however, it does not stand out typologically compared to other languages in terms of indicators of entropy rate or phonological and morphological richness [22–24].

The robustness of Zipf's law under lemmatization for single-author written texts was already checked in previous work studying Spanish, French and English with different

methods [52]. The exponent of lemmas and word forms may vary but both are correlated in texts [52] and the same could happen in speech corpora. However, one could consider an opposite hypotheses that should be confirmed in future works, including more languages: that the lemmatization, as a source of reduction of morphological richness, should vary the exponent of Zipf's rank-frequency law [52], and this variation would depend on the inflectional and derivational complexity of the language (in the sense of Bentz's works [22–24]), that is, it will affect the languages with more morphological variability to a greater extent.

Following [53] the $\alpha$ exponent of Zipf's rank-frequency law "reflects changes in morphological marking" [53], so that more inflection is correlated to a higher $\alpha$ and longer tail of hapax legomena [22, 53]. Comparing modalities, in our case orality shows a lower $\alpha$ than writing, contrary to [53] but consider that here we follow different methods. Besides, it has been shown that there is variability in $\alpha$ related to both the text genre and the linguistic typology for languages of different linguistic families [51]. Future research should control for both effects (modality and morphological complexity) with corpora that are closer and employing uniform methods.

Future work should clarify how factors such as corpus size, binning and linguistic variability, influence Zipf's meaning-frequency law. The relationship between the exponents of the laws confirmed here for Catalan should be investigated for other languages. In any case, other aspects in the formation of words that affect the lemmatization could be considered in future work: the work carried out here for the first time can serve then as a protocol to replicate or refine in future studies with other languages. Our findings call for a revision of previous research of these laws assuming one regime.

## Acknowledgments

We are grateful to C. Bentz, J. M. Garrido, C. Santamaria, J. Rafel and the technicians of Oficines Lexicogràfiques de l'Institut d'Estudis Catalans (Institute of Catalan Studies) for providing us with the data and helpful comments.

## Author Contributions

**Conceptualization:** Neus Català, Jaume Baixeries, Ramon Ferrer-i-Cancho, Lluís Padró, Antoni Hernández-Fernández.

**Formal analysis:** Neus Català, Ramon Ferrer-i-Cancho, Antoni Hernández-Fernández.

**Funding acquisition:** Antoni Hernández-Fernández.

**Investigation:** Neus Català, Ramon Ferrer-i-Cancho, Antoni Hernández-Fernández.

**Methodology:** Neus Català, Ramon Ferrer-i-Cancho, Lluís Padró, Antoni Hernández-Fernández.

**Project administration:** Jaume Baixeries, Antoni Hernández-Fernández.

**Software:** Neus Català, Jaume Baixeries, Lluís Padró.

**Supervision:** Neus Català, Ramon Ferrer-i-Cancho, Antoni Hernández-Fernández.

**Validation:** Neus Català, Ramon Ferrer-i-Cancho, Lluís Padró, Antoni Hernández-Fernández.

**Visualization:** Neus Català, Ramon Ferrer-i-Cancho, Antoni Hernández-Fernández.

**Writing – original draft:** Neus Català, Ramon Ferrer-i-Cancho, Antoni Hernández-Fernández.

**Writing – review & editing:** Neus Català, Ramon Ferrer-i-Cancho, Antoni Hernández-Fernández.

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
