## [Decision Letter · Decision Letter 0]

2 Sep 2021

PONE-D-21-20960

Zipf's laws of meaning in Catalan

PLOS ONE

Dear Dr. Hernández-Fernández,

Thank you for submitting your manuscript to PLOS ONE. After careful consideration, we feel that it has merit but does not fully meet PLOS ONE’s publication criteria as it currently stands. Therefore, we invite you to submit a revised version of the manuscript that addresses the points raised during the review process.

We look forward to receiving your revised manuscript.

Kind regards,

Diego Raphael Amancio

Academic Editor

PLOS ONE

Journal Requirements:

"We are grateful to C. Bentz, J. M. Garrido, C. Santamaria, J. Rafel and the technicians 

of Oficines Lexicogr`afiques de l’Institut d’Estudis Catalans (Institute of Catalan

Studies) for providing us with the data and helpful comments. This work has been 

funded by the projects PRO2020-S03 (RCO03080 Ling¨u´ıstica Quantitativa) and 

PRO2021-S03 HERNANDEZ from Institut d’Estudis Catalans. JB, RFC and AHF are 

also funded by the grant TIN2017-89244-R from Ministerio de Economia, Industria y 

Competitividad (Gobierno de Espa˜na) and supported by the recognition 2017SGR-856

(MACDA) from AGAUR (Generalitat de Catalunya)."

"PRO2020-S03 (RCO03080 Lingüística Quantitativa) from Institut d'Estudis Catalans.

(https://www.iec.cat/)

PRO2021-S03 HERNANDEZ from Institut d'Estudis Catalans.

(https://www.iec.cat/)

JB, RFC and AHF are funded by the grant TIN2017-89244-R from Ministerio de Economia, Industria y Competitividad (Gobierno de España)

(https://www.cnio.es/)

JB, RFC and AHF are supported by the recognition 2017SGR-856 (MACDA) from AGAUR (Generalitat de Catalunya).

(https://agaur.gencat.cat/ca/inici)

The Institut d'Estudis Catalans (https://www.iec.cat/) provided the following datasets: (1) the normative dictionary of the Catalan language (DIEC2), and (2) the written corpus CTILC,"

4. We note that you have stated that you will provide repository information for your data at acceptance. Should your manuscript be accepted for publication, we will hold it until you provide the relevant accession numbers or DOIs necessary to access your data. If you wish to make changes to your Data Availability statement, please describe these changes in your cover letter and we will update your Data Availability statement to reflect the information you provide

Reviewers' comments:

Reviewer's Responses to Questions

**Comments to the Author**

1. Is the manuscript technically sound, and do the data support the conclusions?

Reviewer #1: Yes

Reviewer #2: Yes

2. Has the statistical analysis been performed appropriately and rigorously? 

Reviewer #1: Yes

Reviewer #2: Yes

3. Have the authors made all data underlying the findings in their manuscript fully available?

Reviewer #1: Yes

Reviewer #2: Yes

4. Is the manuscript presented in an intelligible fashion and written in standard English?

Reviewer #1: Yes

Reviewer #2: Yes

5. Review Comments to the Author

Reviewer #1: In this paper, the authors explore the validity of the law of meaning distribution and the meaning-frequency law in Catalan, both in written text and in speech, and argue that such scaling, while present, is better represented by a two-regime (double scaling) than a single scaling one. They also provide rationale for this empirical observation.

The work is definitely interesting and novel, and provides a valuable contribution towards our understanding of the statistical regularities which emerge in language.

I strongly recommend publication, and I would suggest a possible additional statistical experiment which could provide support to the authors findings:

I feel the authors don't provide yet much statistical support for their strong claim (other than visual evidence, which is indeed compelling). Given that they argue that a double scaling is overall a better fit than a single scaling, I would suggest the authors to perform a model selection between two models (single vs double scaling). They should fit the models independently (say, the single scaling via least squares regression, or MLE) and the double scaling via Baayen's method. Once both models are fit, they should be compared statistically. They first would need to compute the likelihood of each of the fits.

The former (single scaling model) has lower complexity (less parameters) while the latter is expected to provide higher goodness of fit (e.g. higher likelihood). A possibility is thus to conclude which model is statistically superior based on some version of AIC (I leave the authors either to take Akaike or some other criterion).

Reviewer #2: The authors present a study of the Zipf’s law of meaning in Catalan. I believe the work presented is novel, of interest to the quantitative linguistics community and beyond, and technically well done. Therefore, I recommend publication of the manuscript with minor corrections. Below, I list some suggestions to help improve the manuscript.

1. Figures could be improved in the following ways: (i) enlarge all labels (ii) increase resolution (higher dpi) (iii) add legends.

2. It is unclear what are the motivations for studying Zipf’s law of meaning in Catalan as opposed to English or other Romance languages. In any case, the authors could discuss whether or not they believe their results can be extended to other languages.

3. The authors should provide access to the code used for the analysis. Although the methods section is quite detailed, the manuscript can not be considered reproducible as it is: the authors should provide more details on the computational aspect (which freeling commands are used? in which order? which flags?), and/or a github repository that allows certain degree of reproducibitliy. If that is not possible or is too complicated, the authors should at least provide some processed data as supplementary data that allows readers to reproduce the main findings of the manuscript in a reasonably simple way.

6. PLOS authors have the option to publish the peer review history of their article (what does this mean?). If published, this will include your full peer review and any attached files.

Reviewer #1: No

Reviewer #2: No

---

## [Author Response · Author response to Decision Letter 0]

12 Oct 2021

Dr. Antoni Hernández-Fernández 

Institut de Ciències de l'Educació. 

Universitat Politècnica de Catalunya 

Jordi Girona 1-3 

08034 Barcelona, Catalonia, Spain 

Editor 

PlosOne Journal.

Dear Editor,

Thank you for giving us the opportunity to submit a revised draft of the manuscript titled Zipf's laws of meaning in Catalan to PlosOne.

We appreciate the effort that you and the reviewers have dedicated to providing valuable feedback on the manuscript. We are grateful to the reviewers for their insightful comments. Regarding the editors' comments, we have detailed the accessibility of the data, corrected typographical errors regarding funding and followed the formal aspects commented by the editors in the first revision.

We also have been able to incorporate the necessary changes to reflect the suggestions provided by the editors and reviewers. We have highlighted the changes within the manuscript in red color. 

Here is a point-by-point response to the editors’ comments and concerns:

 Comments from Editors 

Response: Thank you very much, we ensure about that in this version.

Response: Sorry for this, we ensure about that correct grant numbers are included in this version.

"PRO2020-S03 (RCO03080 Lingüística Quantitativa) from Institut d'Estudis Catalans.

(https://www.iec.cat/)

PRO2021-S03 HERNANDEZ from Institut d'Estudis Catalans.

(https://www.iec.cat/).

JB, RFC and AHF are funded by the grant TIN2017-89244-R from Ministerio de Economia, Industria y Competitividad (Gobierno de España)

(https://www.cnio.es/)

JB, RFC and AHF are supported by the recognition 2017SGR-856 (MACDA) from AGAUR (Generalitat de Catalunya).

(https://agaur.gencat.cat/ca/inici)

The Institut d'Estudis Catalans (https://www.iec.cat/) provided the following datasets: (1) the normative dictionary of the Catalan language (DIEC2), and (2) the written corpus CTILC,"

Response: Thank you very much for this, the previous Funding Statement is correct. 

Response: We include a new version of Data Availability in this new version according to this comment.

Response: Thank you for this comment. We consider it in this new version.

Finally, here is a point-by-point response to the reviewers’ comments and concerns. 

 Comments from Reviewer 1 

Comment 1

The work is definitely interesting and novel, and provides a valuable contribution towards our understanding of the statistical regularities which emerge in language.

Response: Thank you very much. We appreciate the reviewer very much for his/her interest in our paper.

Comment 2

I would suggest a possible additional statistical experiment which could provide support to the authors findings:

I feel the authors don't provide yet much statistical support for their strong claim (other than visual evidence, which is indeed compelling). Given that they argue that a double scaling is overall a better fit than a single scaling, I would suggest the authors to perform a model selection between two models (single vs double scaling). They should fit the models independently (say, the single scaling via least squares regression, or MLE) and the double scaling via Baayen's method. Once both models are fit, they should be compared statistically. They first would need to compute the likelihood of each of the fits.

The former (single scaling model) has lower complexity (less parameters) while the latter is expected to provide higher goodness of fit (e.g. higher likelihood). A possibility is thus to conclude which model is statistically superior based on some version of AIC (I leave the authors either to take Akaike or some other criterion).

Response: Thanks again. We have followed this suggestion in the new version, incorporating in each case both the Akaike Information Criterion with small-sample correction (AICc) and the Bayesian Information Criterion (BIC) calculations for completeness (see Anderson and Burham, 2004), so that new evidence is provided in favor of the two-regime model as a better fit than the one-regime model. Two new sections (Information theoretic model selection) and (Model selection) and three tables with the AICc and BIC results for each Zipfian law (Tables 4, 5 and 6) have been incorporated.

 Comments from Reviewer 2 

Comment 1

I believe the work presented is novel, of interest to the quantitative linguistics community and beyond, and technically well done. 

Response: We appreciate the reviewer very much for his/her interest in our paper.

Comment 2

Figures could be improved in the following ways: (i) enlarge all labels (ii) increase resolution (higher dpi) (iii) add legends.

Response: We have improved the quality of the figures by re-creating them from the beginning and increasing their resolution from 100 dpi to 600 dpi, while keeping dimensions and file sizes in the maximum allowed by this journal. That will also increase the readability of all the labels included in the figures. As for the legends, we have kept them all in the captions for each figure. 

Comment 3

It is unclear what are the motivations for studying Zipf’s law of meaning in Catalan as opposed to English or other Romance languages. In any case, the authors could discuss whether or not they believe their results can be extended to other languages.

Response: Thank you for this comment. In our work we have focused on the study of Catalan because we had an easy access to its dictionary (it was essential to have the number of meanings of each lemma) and to both the oral and written corpora. The study of other languages remains as future work, as indicated in the original version of this article, in the last paragraph:

"Future work should clarify how factors such as corpus size, binning and linguistic variability, influence Zipf’s meaning-frequency law. The relationship between the exponents of the laws confirmed here for Catalan should be investigated for other languages." 

We have added, however, one more sentence about this question, following the reviewer's comment:

In any case, other aspects in the formation of words that affect the lemmatization could be considered in future work: the work carried out here for the first time can serve then as a protocol to replicate or refine in future studies with other languages.

Comment 4

The authors should provide access to the code used for the analysis. Although the methods section is quite detailed, the manuscript can not be considered reproducible as it is: the authors should provide more details on the computational aspect (which freeling commands are used? in which order? which flags?), and/or a github repository that allows certain degree of reproducibitliy. If that is not possible or is too complicated, the authors should at least provide some processed data as supplementary data that allows readers to reproduce the main findings of the manuscript in a reasonably simple way.

Response: Thank you for this comment. We have added a new explanation about Freeling in the main text and the section Data accessibility where we explicitly provide the links to the preprocessed datasets as well as the original sources of those datasets (which were already listed in Table1 in Section Materials in the first version of the paper).

In this section we have also added a link to the configuration file for FreeLing. With these elements, we think that the experiments can be reproduced.

We look forward to hearing from you in due course regarding our submission and to respond to any further questions and comments you may have. 

Best regards,

Dr. Antoni Hernández-Fernández

Universitat Politècnica de Catalunya

References

Anderson, D., Burnham, K. (2004). Model selection and multi-model inference. Berlin: Springer-Verlag.

---

## [Decision Letter · Decision Letter 1]

18 Nov 2021

Zipf's laws of meaning in Catalan

PONE-D-21-20960R1

Dear Dr. Hernández-Fernández,

We’re pleased to inform you that your manuscript has been judged scientifically suitable for publication and will be formally accepted for publication once it meets all outstanding technical requirements.

Kind regards,

Diego Raphael Amancio

Academic Editor

PLOS ONE

Additional Editor Comments (optional):

Reviewers' comments:

Reviewer's Responses to Questions

**Comments to the Author**

1. If the authors have adequately addressed your comments raised in a previous round of review and you feel that this manuscript is now acceptable for publication, you may indicate that here to bypass the “Comments to the Author” section, enter your conflict of interest statement in the “Confidential to Editor” section, and submit your "Accept" recommendation.

Reviewer #1: All comments have been addressed

Reviewer #2: (No Response)

2. Is the manuscript technically sound, and do the data support the conclusions?

Reviewer #1: Yes

Reviewer #2: (No Response)

3. Has the statistical analysis been performed appropriately and rigorously? 

Reviewer #1: Yes

Reviewer #2: (No Response)

4. Have the authors made all data underlying the findings in their manuscript fully available?

Reviewer #1: Yes

Reviewer #2: (No Response)

5. Is the manuscript presented in an intelligible fashion and written in standard English?

Reviewer #1: Yes

Reviewer #2: (No Response)

6. Review Comments to the Author

Reviewer #1: All comments have been addressed. I am happy to recommend publication as is!

Reviewer #2: The authors have addressed all my concerns. I believe the manuscript should be published in its present form.

7. PLOS authors have the option to publish the peer review history of their article (what does this mean?). If published, this will include your full peer review and any attached files.

Reviewer #1: No

Reviewer #2: No

---

## [Editor Report · Acceptance letter]

22 Nov 2021

PONE-D-21-20960R1 

Zipf's laws of meaning in Catalan 

Dear Dr. Hernández-Fernández:

I'm pleased to inform you that your manuscript has been deemed suitable for publication in PLOS ONE. Congratulations! Your manuscript is now with our production department. 

Kind regards, 

on behalf of

Dr. Diego Raphael Amancio 

Academic Editor

PLOS ONE